# A Novel Morphine Drinking Model of Opioid Dependence in Rats

**DOI:** 10.3390/ijms23073874

**Published:** 2022-03-31

**Authors:** Pablo Berríos-Cárcamo, Mauricio Quezada, Daniela Santapau, Paola Morales, Belén Olivares, Carolina Ponce, Alba Ávila, Cristian De Gregorio, Marcelo Ezquer, María Elena Quintanilla, Mario Herrera-Marschitz, Yedy Israel, Fernando Ezquer

**Affiliations:** 1Center for Regenerative Medicine, Faculty of Medicine Clínica Alemana, Universidad del Desarrollo, Santiago 7610658, Chile; pablo.berrios@udd.cl (P.B.-C.); mquezadad@udd.cl (M.Q.); dsantapau@udd.cl (D.S.); aavilas@udd.cl (A.Á.); crdegregorio@udd.cl (C.D.G.); mezquer@udd.cl (M.E.); yisrael@uchile.cl (Y.I.); 2Molecular and Clinical Pharmacology Program, Institute of Biomedical Science, Faculty of Medicine, Universidad de Chile, Santiago 8380453, Chile; pmorales@uchile.cl (P.M.); equintanilla@uchile.cl (M.E.Q.); mh-marschitz@med.uchile.cl (M.H.-M.); 3Department of Neuroscience, Faculty of Medicine, Universidad de Chile, Santiago 8380453, Chile; 4Research Center for the Development of Novel Therapeutic Alternatives for Alcohol Use Disorders, Santiago 8900000, Chile; 5Center for Medical Chemistry, Faculty of Medicine Clínica Alemana, Universidad del Desarrollo, Santiago 7610658, Chile; molivares@udd.cl; 6Faculty of Agricultural and Forestry Sciences, Universidad de la Frontera, Temuco 4811230, Chile; c.ponce03@ufromail.cl

**Keywords:** morphine, opioids, addiction, oral intake, animal model, dependence, quinine

## Abstract

An animal model of voluntary oral morphine consumption would allow for a pre-clinical evaluation of new treatments aimed at reducing opioid intake in humans. However, the main limitation of oral morphine consumption in rodents is its bitter taste, which is strongly aversive. Taste aversion is often overcome by the use of adulterants, such as sweeteners, to conceal morphine taste or bitterants in the alternative bottle to equalize aversion. However, the adulterants’ presence is the cause for consumption choice and, upon removal, the preference for morphine is not preserved. Thus, current animal models are not suitable to study treatments aimed at reducing consumption elicited by morphine itself. Since taste preference is a learned behavior, just-weaned rats were trained to accept a bitter taste, adding the bitterant quinine to their drinking water for one week. The latter was followed by allowing the choice of quinine or morphine (0.15 mg/mL) solutions for two weeks. Then, quinine was removed, and the preference for morphine against water was evaluated. Using this paradigm, we show that rats highly preferred the consumption of morphine over water, reaching a voluntary morphine intake of 15 mg/kg/day. Morphine consumption led to significant analgesia and hyperlocomotion, and to a marked deprivation syndrome following the administration of the opioid antagonist naloxone. Voluntary morphine consumption was also shown to generate brain oxidative stress and neuroinflammation, signs associated with opioid dependence development. We present a robust two-bottle choice animal model of oral morphine self-administration for the evaluation of therapeutic interventions for the treatment of morphine dependence.

## 1. Introduction

Opioids are widely used to treat acute and chronic pain due to their potent analgesic and sedative properties. However, since opioids also generate the compulsion to take the drug, it has led to widespread non-medical opioid use [1].

Opioid drugs are among the most addictive substances [1]. Repeated daily use leads to physical dependence and rapid tolerance to their effects [1]. Thus, opioid dependence has become an epidemic concern, with over 16 million opioid-dependent individuals worldwide and over 130,000 deaths reported annually [2], representing 76% of deaths in which a drug-use disorder was implicated [3,4].

It is consistently reported that the oral route is the preferred route of administration for prescription opioids. Several studies show that over 90% of abusers of prescription opioids, including oxycodone, hydrocodone, and morphine, report oral ingestion for non-medical purposes [5,6]. Thus, the generation of animal models of voluntary opioid oral intake has become a necessary pre-clinical tool to evaluate new pharmacological strategies for the treatment of opioid dependence.

Studies evaluating the voluntary oral self-administration of drugs of abuse in animal models often utilize procedures in which the preference for drug-containing solutions is assessed against either water or palatable non-drug solutions. Among the different paradigms commonly used, the two-bottle free-choice exposure is widely used to evaluate drug avidity, being a non-invasive procedure in which animals have the option of voluntarily drinking the desired amount of drug. This paradigm has demonstrated relevant translational value for several drugs of abuse [7,8]. However, the main limitation of oral morphine intake is its bitter taste. Thus, when the morphine bottle is paired with a bottle containing non-adulterated tap water, animals prefer the non-morphine solution [9]. To overcome this limitation, researchers reported the addition of taste adulterants such as sweeteners, e.g., saccharin, added to the morphine solution or bitterants, e.g., quinine, in the alternative bottle to increase morphine consumption and to demonstrate morphine dependence [9,10]. However, the neurobiology by which quinine and saccharin influence the ensuing preference for morphine in rodent models remains poorly understood.

Ferraro et al. reported that the addition of 0.2% saccharin to the morphine solution could mask the bitter taste of morphine, increasing morphine preference compared to the control bottle [11]. However, morphine and saccharin have known pharmacological interactions since saccharin potentiates morphine-induced antinociception in mice [12]. Furthermore, saccharin consumption is also known to promote dopamine release in the nucleus accumbens, activating the mesolimbic reward system [13]. Even after several days of high intake of morphine plus saccharin, the elimination of saccharin from the bottles markedly reduces morphine consumption [11], thus limiting the translational value of this model. Similarly, rats given morphine adultered with sugar in their drinking water for 21 days do not develop a preference for the morphine-sugar bottle when evaluated against a bottle with only sugar dissolved in water [14]. Thus, the taste-altering confounding factor must be eliminated before animals that consume morphine could be reliably used for pre-clinical testing of treatments aimed at reducing consumption elicited by morphine itself.

On the other hand, when quinine is included in the alternative bottle, a preference for the morphine solution is observed, suggesting that quinine is effective in driving morphine consumption and generating dependence. However, several reports demonstrated that, in this paradigm, high morphine preference is not related to the rewarding properties of the drug, but its preference is influenced by the degree of quinine aversion [15,16]. Thus, it was postulated that the aversion to the bitter taste of quinine is what leads to the maintenance of relatively high levels of fluid consumption from the morphine bottle [15]. This interpretation is supported by the significant reduction in morphine intake observed when quinine is eliminated from the alternative bottle, despite weeks of morphine consumption [11,15]. Thus, new paradigms to induce a preference for morphine-containing solutions without the influence of taste adulterant confounders are highly needed.

Rodents, similar to humans, are born with only a few innate flavor preferences and aversions, and they exhibit marked plasticity throughout the individual’s lifespan, being highly sensitive to modification by experience [17]. In rodents, flavor preference learning usually occurs at the time of weaning and could influence food selection after weaning [17]. Thus, repeated exposure to an aversive flavor early in life could improve flavor acceptance in adult life due to familiarity, reducing inherent neophobia [18]. The bitter taste is one of these aversive flavors. Interestingly, exposure of young rats to the bitterant quinine as their only source of fluid immediately after weaning significantly increases quinine preference when evaluated in a two-bottle choice against water [19]. Those results suggest that quinine consumption after weaning makes quinine, and possibly other bitterants, more palatable to weanling rats.

In this study, we show a novel morphine drinking paradigm in which, immediately after weaning, rats are exposed for seven days to a quinine solution as their only fluid source, to make animals get used to a bitter taste. After that period, animals are exposed to a two-bottle choice of quinine or morphine solutions for two weeks. Finally, the quinine bottle is replaced by a water bottle, and animals now have the choice of drinking morphine or water. After four weeks of voluntary morphine intake, we evaluated classical markers of the pharmacological action of morphine, including thermal and mechanical analgesia and hyperlocomotion, as well as markers of morphine dependence including the appearance of somatic signs of morphine withdrawal triggered by the administration of naloxone, an opioid receptor antagonist. In additional studies, we assessed other molecular markers of opioid-induced dependence, including morphine-induced neuroinflammation and oxidative stress and the mRNA levels of µ-opioid receptor and accessory proteins.

## 2. Results

### 2.1. Morphine Oral Consumption

Animals were trained at a young age to accept the bitter taste of quinine (*n* = 29). Upon weaning, three-week-old rats were exposed to a quinine solution (0.15 mg/mL) as their only drinking fluid for seven days. Despite the high quinine concentration, which is shown to produce significant suppression of fluid and food intake in adult rats [19], the consumption of quinine solution following the first two days of access was not significantly different from that of the water-only exposed animals (*n* = 8) (Figure 1A), and their body weight gains were indistinguishably between both groups (Figure 1B).

Following seven days of quinine exposure, a morphine-solution (0.15 mg/mL) containing bottle was added in addition to the quinine-containing bottle (a water-only bottle was not available). A marked preference for the novel morphine solution was quickly observed, and the animals reached a ~90% preference for the morphine solution over the quinine solution after five days of the two-bottle choice for the bitter solutions (Figure 2A,B). Only four animals (14%) kept a significant quinine preference (below 50% preference for the morphine solution) and were removed from the study. While a modest downward intake of the morphine solution was seen thereafter, likely the result of the fast increases in body weight of recently weaned animals, the high preference for the morphine solution remained, which was consistent with a reduced total fluid consumption normalized by body weight (Figure 2C,D).

After two weeks of free-choice between morphine and quinine solutions, the quinine bottle was replaced by one filled with only water to assess the morphine consumption without the confounding presence of quinine. Despite a small initial drop in preference for morphine, the preference for morphine stayed higher than 80% and, importantly, voluntary morphine intake stayed above 15 mg/kg/day for the remaining two weeks of observation (Figure 2A), reaching morphine plasma levels of 2.5 ± 0.6 mg/mL measured during the first hour of light, to expect the highest plasmatic morphine level, since rats mainly drink overnight. It is noted that seven animals changed their preference ratio from the morphine solution to water during the two weeks of observation and were removed from the study. Thus, the overall rejection corresponds to 11 animals of the original 29 (38%).

### 2.2. Orally Consumed Morphine Was Pharmacologically Relevant

To determine whether rats self-administering 15–18 mg morphine/kg/day experienced the pharmacological effects of morphine, we determined if the animals (i) showed morphine-induced analgesia, (ii) showed morphine-induced hyperlocomotion, and (iii) developed a physical dependence shown by somatic signs consistent with the opioid withdrawal syndrome, after the systemic administration of the opioid receptor antagonist naloxone [20,21].

*(i) Morphine-induced analgesia*. As a measure of the analgesic effect of morphine, rats that had voluntarily consumed morphine for four weeks were tested for their withdrawal latency following a thermal stimulus and for the withdrawal threshold to a mechanical stimulus. Morphine-consuming animals showed a significantly higher thermal withdrawal latency (Figure 3A) and a significantly higher mechanical withdrawal threshold (Figure 3B) compared to water-drinking controls (*p* < 0.0001).

*(ii) Morphine-induced hyperlocomotion.* Rats that had consumed morphine and were placed on an open field showed a higher distance traveled in five minutes (*p* < 0.001) compared to water-drinking rats (Figure 4A). The experimental rats also spent a lower percentage of the session close to the compartment walls, decreasing thigmotaxis compared to control rats (*p* < 0.05) (Figure 4B,C), a measurement of anxiety reduced by morphine treatment [22].

*(iii) Morphine withdrawal syndrome*. Animals that had voluntarily consumed morphine also developed physical dependence. This was shown by the eliciting of withdrawal syndrome signs following the administration of the opioid antagonist naloxone. Specifically, morphine-consuming rats showed diarrhea and a significant weight loss (*p* < 0.0001) 30 min after the naloxone injection (Figure 5A,B). Animals also expressed more often the behavioral signs associated with morphine withdrawal, such as wet-dog shakes, forepaw tremors, and chewing (*p* < 0.05) (Figure 5C–E). Additionally, animals showed reduced rearing (*p* < 0.001) during the 30 min observation period after the naloxone injection (Figure 5F), a behavior that is described as a withdrawal sign of orally consumed morphine [20].

### 2.3. Brain Changes in Rats That Voluntarily Consumed Morphine

To study whether rats that had voluntarily consumed morphine displayed brain neurochemical alterations, we determined the presence of neuroinflammation, oxidative stress levels, and morphine-related protein expression. *Hippocampal neuroinflammation:* Neuroinflammation was evaluated by determining glial reactivity in the hippocampus, assessed as morphological changes in astrocytes and microglial density. Data show that a time period of four weeks of voluntary morphine intake induces a significant increase (*p* < 0.0001) in the length and thickness of hippocampal astrocyte processes (GFAP immunofluorescence) versus those of control animals drinking water (Figure 6A,B). Chronic morphine intake also led to a significant increase (*p* < 0.001) in microglial density (Iba-1 immunofluorescence) compared to control animals (Figure 6A,C).

*Hippocampal oxidative stress levels*. Oxidized glutathione (GSSG)/reduced glutathione (GSH) levels were assessed in the hippocampus of rats that consumed morphine voluntarily and proper controls to determine the GSSG/GSH ratio, as a validated marker of oxidative stress [23,24]. Data show a significant five-fold increase (*p* < 0.001) in the GSSG/GSH ratio in the hippocampus of morphine-consuming rats compared to that of water-drinking rats (Figure 7).

*Morphine-related signaling pathway*. We observed a significant (*p* < 0.01) increase in the levels of µ-opioid receptor mRNA transcript 1 (OPRM1) in the pre-frontal cortex of rats that consumed morphine voluntarily compared to water-drinking rats (Figure 8). A significant increase (*p* < 0.05) in the pre-frontal cortex was also observed for the mRNA of the accessory protein Regulator of G Protein Signaling 17 (RGS17), but not for the accessory protein vasoactive intestinal peptide (VIP), which are proteins responsible for the modulation of opioid signaling [25,26] (Figure 8). On the other hand, no differences were observed in the nucleus accumbens in the mRNA levels of these proteins, suggesting that the regulation in the mRNA levels of OPRM1 and RGS17 is region-specific.

## 3. Discussion

Opioid dependence is mostly initiated by the oral administration of prescription opioids, and over 90% of abusers of prescription opioids report their oral ingestion for non-medical purposes [5,6]. While several models of intravenous opioid self-administration are reported [27,28], these do not fully reflect the changes that result from the pharmacokinetic of oral opioid administration. Thus, the generation of a novel and reliable animal model of voluntary oral opioid intake resulting in marked dependence, as shown by a clear naloxone-induced precipitated withdrawal, would constitute a valuable pre-clinical tool to evaluate new pharmacological strategies for the treatment of oral opioid dependence.

The main limitation in the development of an animal model of oral morphine intake is its bitter taste. Thus, the aim of the present study was to train rats to accept the taste of morphine. In rodents, learning taste preference occurs primarily at weaning time and can influence food selection later in life [17]. It was postulated that repeated exposure to a known aversive flavor at a young age could improve animal flavor acceptance in adult life [18]. Indeed, exposure of rats immediately after weaning to the bitter taste of a quinine solution as the only source of fluid for seven days significantly increases subsequent quinine preference [19]. This was confirmed in the present study since, after only two days of access to a quinine solution as the only source of fluid, rats ingested the quinine solution at the same rate as rats that were allowed water access only. Following seven days of quinine solution intake, rats were able to drink both quinine and morphine solutions, while preferring the morphine solution and further preferring morphine over water, reaching an 80% preference for the morphine solution versus water. These findings suggest that exposure to quinine after weaning renders morphine more palatable.

Although we did not explore the pharmacological effect of quinine consumption during the seven-days period of exclusive consumption of just-weaned rats, we do not expect persisting alterations that may confound the findings associated with the morphine consumption. In that period, rats consumed quinine 30 mg/kg/day, which is comparable to the dose used as an anti-malarian drug in humans. In patients following that regime, mild and reversible adverse reactions are common. Cichonism, observed in most patients treated with quinine 30 mg/kg/day, is comprised of nausea, headache, tinnitus, hearing impairment, and blurred vision [29]. In our study, morphine consumption against water was analyzed 15 to 28 days after rats were drinking high doses of quinine, ensuring the disappearance of the possible adverse effects of quinine exposure.

A limitation of our model is the number of subjects required to be removed from the study. Because our goal was to develop a robust model that could be used to study therapeutics to reduce morphine consumption of dependent subjects, we rejected animals that did not show a preference for morphine against the alternative bottle. Of those rejected, 13% (four animals) were removed after the morphine bottle was given to the rats, which consumed quinine exclusively until then. This suggests that the morphine and quinine solutions were similarly aversive, representing a preference before the animals became dependent. Seven additional rats were removed from the study when the quinine bottle was replaced by one containing only water, since those animals changed their preference from morphine to water. Thus, 18 rats (62% of the starting group of rats) preferred morphine for the complete extension of the study. These rejection rates are in concordance to what was previously reported for other drugs since a diversity in the willingness to consume a drug is observed when outbred animals, i.e., not selectively bred to self-administer a drug, are used as a model for the study of consumption of ethanol [30,31].

Another limitation of our model is the specific age requirement for starting experiments, that being at weaning age, which results in the experiments being performed using young rats. Animals were eight weeks old when euthanasia was performed. On the other hand, our model could be useful for the study of morphine consumption in young individuals, an age segment with high prevalence. In humans, the average age of opioid initiation is between 22 and 23 years old, and teenagers report high availability of illicit opioids [32].

Other research groups also show that morphine is voluntarily consumed by animals when quinine is given in the alternative bottle. However, they show a marked reduction in morphine consumption when quinine is reduced or removed from the alternative bottle [11,15], which suggests that the rewarding/reinforcing effect of morphine (and an induced dependence) was not the main cause of their increased intake.

In the present study, we show that after four weeks of voluntary morphine intake (versus water in controls), the classical acute effects and chronic dependence on morphine effects are observed, including: (i) thermal and mechanical analgesia; (ii) morphine-induced hyperlocomotion, and importantly (iii) naloxone precipitated withdrawal. The later included (a) significant diarrhea, (b) marked weight loss, (c) wet-dog shakes, (d) forepaw tremors, (e) chewing, and (f) reduced rearing, as previously reported [20,21].

Other studies also aimed at developing animal models for oral opioid self-administration. It is shown that mice [33,34] and rats [35] voluntarily consume oxycodone. In studies in mice, a period of forced oxycodone intake was initially imposed under post-prandial conditions (in the early dark cycle) which was followed by operant self-administration oxycodone intake conditioned by light and tones. In the studies using rats, the animals were initially water deprived for 22 h a day and exposed to oxycodone solutions as their only fluid, also followed by operant self-administration. While, in all these studies, the animals learned to self-administer oxycodone orally, a naloxone precipitated withdrawal was not reported under these oral self-administration conditions. Indeed, in studies using mice, authors induced the naloxone precipitated withdrawal in animals that were administered oxycodone systemically, rather than orally. In contrast, in the rat study of oral operant self-administration of oxycodone, a high dose of naltrexone was not reported to generate the signs of opioid withdrawal. Data suggests that means to induce adult animals to consume oxycodone orally do not readily result in consumption levels that lead to the development of clear physical dependence.

In the present studies, molecular markers of opioid-induced dependence were examined, including higher levels of µ-opioid receptor and RGS17 mRNA levels in the pre-frontal cortex. The latter is a regulator of G-protein signaling that directly interacts with the µ-opioid receptor and promotes opioid tolerance [36]. Increased levels of RGS17 mRNA are associated with morphine preference [36,37]. These alterations were not observed in the nucleus accumbens, suggesting that they are region-specific.

An animal model for voluntary morphine consumption should also reproduce relevant molecular and histological brain alterations that are commonly observed in human patients after morphine exposure. Morphine treatment is shown to promote a rise in brain oxidative stress by both a direct increase of free radicals’ levels and a reduction in the enzymatic antioxidant machinery [38], and increases neuroinflammation by the direct activation of Toll-like receptors [39], the hippocampus being the most affected brain region. These phenomena are shown to characterize opioid tolerance and may represent the engines of relapse [38,40]. Indeed, the administrations of antioxidant and anti-inflammatory agents are shown markedly to reduce morphine withdrawal [41] and conditioned place preference [42]. In general, antioxidant and anti-inflammatory agents are shown to reduce the self-administration and relapse of many drugs of abuse, supporting the hypothesis that brain oxidative stress and neuroinflammation are required for an addiction to develop [43]. In the present oral morphine intake model, molecular markers of morphine-induced dependence were clearly observed, including a significant increase in morphine-induced neuroinflammation evidenced by an increase in the length and thickness of primary astrocytic processes and an increase in microglial density in the hippocampus. As expected, morphine-induced neuroinflammation was also accompanied by a significant increase in brain oxidative stress evidenced by an elevated GSSG/GSH ratio in the hippocampus.

### Conclusions

Overall, studies conducted show a novel two-bottle choice paradigm to induce a preference for morphine-containing solutions, that starts with young weaning rats, with no influence of taste adulterant confounders nor the need for the animals to consume it to avoid an externally imposed punitive condition. These animals voluntarily drink morphine-containing solutions in amounts that lead to clear morphine dependence, as seen by the classical naloxone-induced signs, and display the molecular and histological brain alterations commonly observed in opioid addicted patients. Thus, this animal model could be valuable for the testing of new therapeutic interventions for the treatment of this devastating and extending condition.

## 4. Materials and Methods

### 4.1. Animals

Three-week-old female Wistar rats were used in the experiments. Female rats were chosen since it is reported that female rats consume or self-administer higher levels of morphine than male rats [44,45,46]. Rats were single-housed at a constant temperature and humidity, on a 12 h light/dark (normal) cycle and with unrestricted access to standard chow and environmental enrichment. All animal procedures were approved by the Committee for Experiments with Laboratory Animals of the Faculty of Medicine of the Universidad del Desarrollo (Protocol 05/2020).

### 4.2. Voluntary Morphine Consumption Test: Free-Choice Drinking Paradigm

Immediately after weaning, three-week-old rats were randomly assigned to one of two groups. One group was exposed to 0.15 mg/mL quinine hydrochloride (Sigma-Aldrich, St. Louis, MO, USA) dissolved in tap water as their only fluid for consumption to accustom the animals to a bitter taste (*n* = 29) [19]. Animals in the second group were only exposed to water (control group) (*n* = 8). Quinine intake, water intake, and body weight were recorded daily. Seven days after starting quinine exposure, animals were changed for two weeks to a two-bottle choice paradigm in which one bottle contained 0.15 mg/mL quinine hydrochloride dissolved in tap water, and the other bottle contained the bitter 0.15 mg/mL morphine sulfate (Oramoph; Molteni Farmaceutici, Scandicci, Florence, Italy) dissolved in tap water. Thereafter, the quinine bottle was removed, and animals were exposed to 0.15 mg/mL morphine sulfate and water (two-bottle choice) for two additional weeks. Animals drinking only water were maintained as controls. Morphine intake, morphine preference, total fluid consumption, and animal weights were measured daily (Appendix A). For two-bottle choice tests, the position of each bottle was alternated daily to prevent the accustoming of the animal to drinking from the bottle of a specific position.

*Criteria for subject removal from the study*: To develop a robust model to study therapeutics that could reduce morphine consumption, we defined the minimal morphine preference over the alternative bottle to be >50% in volume consumed. Thus, animals that kept a morphine preference below 50% for two consecutive days were removed. There were two instances when subjects were removed: (i) when a morphine bottle was presented to rats drinking only quinine and (ii) when the quinine bottle was replaced with a water bottle.

### 4.3. Plasma Morphine Determination

After four weeks of voluntary morphine intake, rats were anesthetized by inhalation of 4% sevoflurane vapor (Baxter, Deerfield, IL, USA) in oxygen, and blood samples were collected by cardiac puncture during the first hour of light to expect the highest plasmatic morphine level, since rats mainly drink overnight. Morphine level was determined in the plasma by ELISA using the Morphine ELISA kit (Cayman Chemical, Ann Arbor, MI, USA) following the manufacturer’s instructions. Methanol extracted samples were vacuum dried, resuspended in the provided buffer, and 1000-fold diluted before the analysis.

### 4.4. Evaluation of Morphine-Induced Analgesia

Following four weeks of voluntary morphine intake, analgesia was assessed by the determination of thermal and mechanical sensitivity. For thermal sensitivity determinations, the rats were placed for 20 min inside an acrylic box provided with a mobile on-off infrared light (Ugo Basile Plantar Test) below the surface of the box. Thereafter, the infrared light was placed beneath the mid-plantar surface of a hind paw, and the paw withdrawal response was automatically recorded. The maximum exposure was set at 15 s to avoid hind paw damage. The infrared light exposure was performed to either paw until completing three measurements, with an interval of five minutes between stimuli, and measurements were repeated the following day. The complete control group (*n* = 8) and a random sample of 8 animals from the morphine consumption group were assayed. Data in seconds were expressed as the mean withdrawal latency registered each day as we previously reported [47].

For mechanical sensitivity determination, animals were placed inside an acrylic box with a mesh floor that allows free access to the plantar surface of the paw. Then, the mid-plantar surface of one of the hind paws was stimulated with an electronic Von Frey filament (Electronic Von Frey; Ugo Basile, Comerio, Varese, Italy) with increasing strength, and the paw withdrawal response was automatically recorded. The stimulation was performed to either paw until completing three measurements, with an interval of five minutes between stimuli, and measurements were repeated the following day. The complete control group (*n* = 8) and a random sample of 8 animals from the morphine consumption group were assayed. Thermal and mechanical sensitivities were measured in a 23 °C controlled temperature room during the first hour of light to expect the highest plasmatic morphine levels.

### 4.5. Evaluation of Morphine-Induced Locomotor Activity

Following four weeks of voluntary morphine intake, locomotor activity was assessed by the open field test. The open field consisted of a square base (100 × 100 cm) and walls of 40 cm made of black painted wood. Rats were placed in the center, and their locomotor activity was recorded for five minutes with a video camera placed in a zenithal position. The open field arena was cleaned with 70% ethanol after each animal session. The room was lit by a soft white light (50–60 lux). The experiments were performed during the first hour of light to expect the highest plasmatic morphine levels. Total distance traveled and thigmotaxis time (tendency to remain close to the walls) were evaluated using the ANY-maze video tracking system (ver. 6.35, Stoelting Co., Wood Dale, IL, USA). The complete control group (*n* = 8) and a random sample of 8 animals from the morphine consumption group were assayed.

### 4.6. Induction of Withdrawal Syndrome

For induction of the withdrawal syndrome, rats that had voluntarily ingested the morphine solution for four weeks were intraperitoneally injected 5 mg/kg of the opioid receptor antagonist naloxone (Sigma-Aldrich) dissolved in 0.9% saline. Immediately after naloxone administration, animals were placed in a glass beaker (300 mm height and 180mm diameter) and monitored for weight loss, diarrhea, wet-dog shakes, forepaw tremors, chewing, and rearing events over 30 min. The complete control group (*n* = 8) and a random sample of 8 animals from the morphine consumption group were assayed.

### 4.7. Evaluation of Morphine-Induced Neuroinflammation

Neuroinflammation was evaluated, determining astrocyte activation and microglial density in the hippocampus of rats after four weeks of voluntary morphine intake. Animals were anesthetized by inhalation of 4% sevoflurane vapor in oxygen (Baxter), intracardially perfused with 100 mL of PBS (pH 7.4), and euthanized to obtain brain samples. Double-labeling immunofluorescence against the astrocyte marker glial fibrillary acidic protein (GFAP) and the microglial marker ionized-calcium-binding adaptor molecule 1 (Iba-1) were evaluated in coronal 30 mm thick cryo-sections of hippocampus as we previously reported [48]. Briefly, coronal sections were washed with 0.1 M PBS, incubated with blocking solution (0.3% Triton X-100, 0.1% BSA, and 10% normal goat serum in PBS) for 1 h, followed by incubation with a primary rabbit monoclonal anti-IBA-1 antibody (cat#019-19741, Wako, 1:500 dilution in blocking solution) at 4 °C overnight. Thereafter, sections were rinsed in PBS containing 0.3% Triton X-100 and incubated with goat anti-rabbit secondary antibody (Alexa Fluor 594, Thermo Fisher Scientific, Waltham, MA, USA; 1:500 dilution in blocking solution) for 1 h in the dark at room temperature. Then, sections were rinsed, incubated in blocking solution (0.3% Triton X-100, 1% BSA, and 5% normal goat serum in PBS) for 1 h and subsequently with a primary mouse monoclonal anti-GFAP antibody (cat#G3893, Sigma-Aldrich, 1:500 dilution in blocking solution) overnight at 4 °C in the dark. Samples were rinsed with PBS, incubated with goat anti-mouse secondary antibody (Alexa Fluor 488, Thermo Fisher Scientific, 1:500 dilution in blocking solution) for 1 h in the dark at room temperature. Nuclei were counterstained with DAPI (Invitrogen, Waltham, MA, USA; 0.02M: 0.0125 mg/mL). Microphotographs were taken from the stratum radiatum of the hippocampus using a confocal microscope (Olympus FV10i). The area analyzed for each stack was 0.04 mm^2^, and the thickness (Z axis) was measured for each case. The total length and thickness of the GFAP-positive astrocyte primary process and density of Iba-1-positive microglial cells were assessed using FIJI image analysis software as previously reported [24]. The complete control group (*n* = 8) and a random sample of 8 animals from the morphine consumption group were assayed.

### 4.8. Evaluation of Morphine-Induced Oxidative Stress

Brain oxidative stress was evaluated, determining the oxidized glutathione (GSSG)/reduced glutathione (GSH) ratio (GSSG/GSH) in the hippocampus of rats after four weeks of voluntary morphine intake. To this end, the hippocampi were extracted and mixed with three volumes of ice-cold potassium buffer (0.1M) containing 5 mM EDTA, pH 7.4. GSH and GSSG contents were determined as previously described [24]. Briefly, GSSG in the sample was first converted into GSH with glutathione reductase (Sigma-Aldrich) and NADPH (Sigma-Aldrich). The total free thiol group of GSH was reacted with the sulfhydryl reagent DTNB (Sigma-Aldrich), yielding a product that absorbs light at 412nm. GSSG per se in the homogenate was measured by adding 2-vinyl pyridine (Sigma-Aldrich) to trap GSH, preventing GSH from binding DTNB. The excess of 2-vinyl pyridine was neutralized with triethanolamine. Finally, the free GSH levels were obtained by subtracting the GSSG value to total GSH. The complete control group (*n* = 8) and a random sample of 8 animals from the morphine consumption group were assayed.

### 4.9. Determination of mRNA Levels of Morphine Receptor and Accessory Proteins

Four weeks after voluntary morphine intake, total RNA was isolated from the nucleus accumbens and pre-frontal cortex using TRIzol reagent (Invitrogen) following manufacturer’s instructions. One microgram of total RNA was used to perform reverse transcription with MMLV reverse transcriptase (Invitrogen) and oligo dT primers. Real-time PCR reactions were performed in a 10 mL final volume containing: 50ng cDNA, PCR LightCycler-DNA Master SYBERGreen reaction mix (Roche, Basel, Switzerland), 3 mM MgCl_2_ and 0.5mM of the primers for the amplification of the opioid receptor OPRM1, and the accessory proteins RGS17 and VIP using a Light-Cycler 1.5 thermocycler (Roche) as previously reported [23]. To ensure that amplicons were generated from mRNA and not from genomic DNA, controls without RT during the reverse transcription reactions were included. Expression levels were determined by the ∆∆Ct method. The mRNA level of each target gene was normalized against the mRNA levels of the housekeeping genes glyceraldehyde 3-phosphate dehydrogenase (GAPDH) and hypoxanthine phosphoribosyltransferase 1 (Hprt1) in the same sample. The primers used for qPCR reactions were: OPRM1 forward 5′-ATCCAGTTCTTTACGCCTTCC-3′; OPRM1 reverse 5′- GATGTTCCCTAGTGTTCTGA CG-3′; RGS17 forward 5′-GAACAGAATACAGCGAGGAGAA-3′; RGS17 reverse 5′-GACCTCTTTCGGTGACAGTATAG-3′; VIP forward 5′-CATTGGCAAACGAATCAGCA GT-3′; VIP reverse 5′- CTCACTGCTCCTCTTCCCATTTAG-3′; GAPDH forward 5′-GACATGCCGCCTGGAGAAAC-3′; GAPDH reverse 5′-AGCCCAGGATGCCCTTTAGT-3′; Hprt1 forward 5′-CTGGTGAAAAGGACCTCTCG-3′; Hprt1 reverse 5′-TCCACTTTCGCTGATGACAC-3′. The complete control group (*n* = 8) and a random sample of 8 animals from the morphine consumption group were assayed.

### 4.10. Statistical Analysis

Statistical analyses were performed using GraphPad Prism software. Data are expressed as mean ± SEM. Two-way (treatment × day) analysis of variance (ANOVA), followed by the Bonferroni post hoc test, was conducted to compare liquid intake. Student’s *t*-test was used to determine if two sets of data were significantly different from each other. A level of *p* < 0.05 was considered for statistical significance. GraphPad Prism version 9.2.0 was used for statistical analysis. To facilitate text reading, the full statistical analyses are presented in the figure legends.

## Figures and Tables

**Figure 1 ijms-23-03874-f001:**
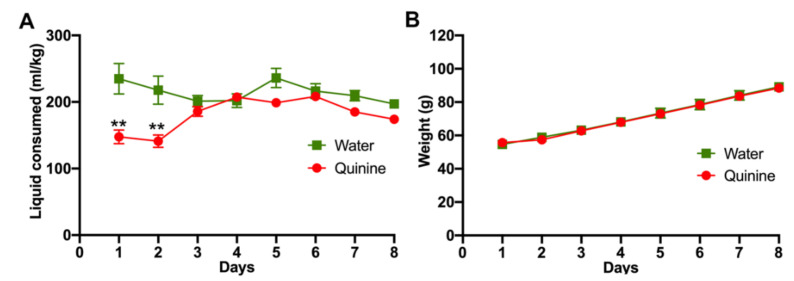
Young rats learn to drink a bitter-tasting quinine solution. Immediately after weaning, three-week-old female Wistar rats were exposed to water or 0.15 mg/mL quinine dissolved in water as the only fluid choice. (**A**) Water or quinine solution intake was evaluated daily and expressed as ml consumed/kg body weight. (**B**) Animal weights were determined daily. *n* = 8 in the water group and *n* = 29 in the quinine group. Data are expressed as mean ± SEM. ** *p* < 0.01, Two-way ANOVA.

**Figure 2 ijms-23-03874-f002:**
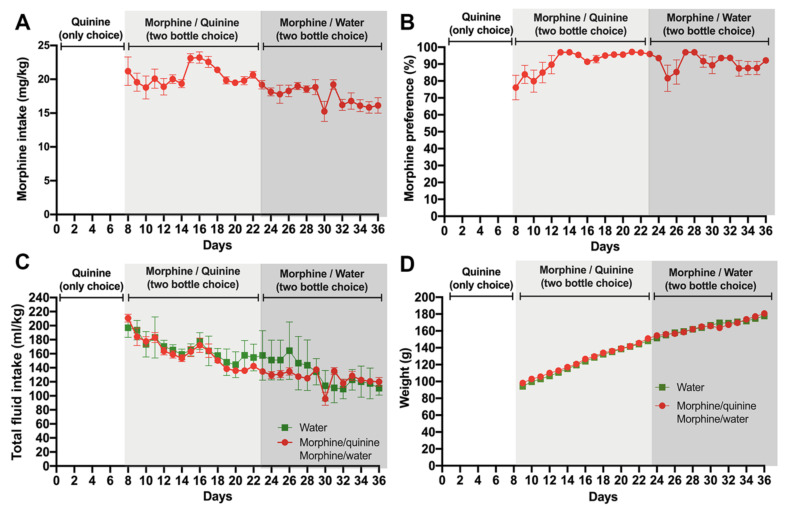
Voluntary morphine intake by Wistar rats. One week after starting quinine intake (Figure 1), animals were exposed to 0.15 mg/mL quinine and 0.15 mg/mL morphine for two weeks using the two-bottle choice paradigm. Thereafter, animals were exposed to 0.15 mg/mL morphine and water (two-bottle choice) for two additional weeks. (**A**) Voluntary morphine intake is expressed as mg morphine consumed per kg of body weight per day. (**B**) Morphine preference (%). (**C**) Total fluid intake expressed as total fluid consumed per kg of body weight/day. (**D**) Animal weights. Data are expressed as mean ± SEM. *n* = 8 in the water group, and *n* = 18 in the morphine group; 38% of the animals in the morphine group were excluded since they did not show >50% morphine preference against water.

**Figure 3 ijms-23-03874-f003:**
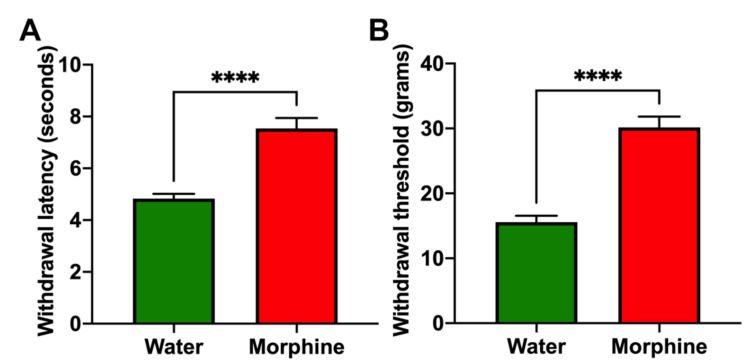
Voluntary morphine intake induces analgesia in Wistar rats. Hargreaves plantar test (**A**) and Von Frey test (**B**) showing the withdrawal latency (delayed response) to thermal stimulus and withdrawal threshold to mechanical stimulus, respectively, of rats that had voluntarily ingested a morphine solution for four weeks. Animals drinking only water were used as controls. Withdrawal latency and withdrawal threshold were measured during the first hour of light. Data are expressed as mean ± SEM. *n* = 8 in each experimental group. **** *p* < 0.0001, two-tailed Student *t*-test.

**Figure 4 ijms-23-03874-f004:**
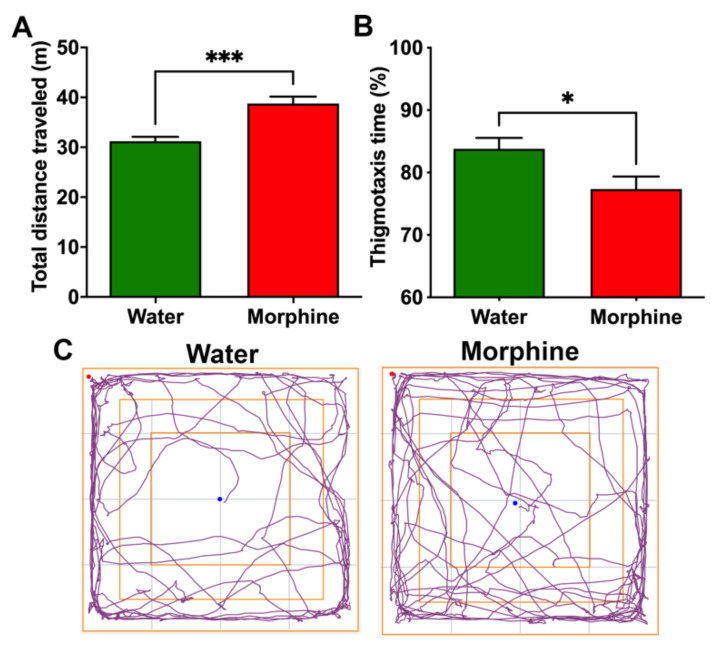
Voluntary morphine intake induces hyperlocomotor activity in Wistar rats. (**A**) Total distance traveled and (**B**) Thigmotaxis time (%) as a measure of time spent near the walls during the five minute session of the open field test of rats that had voluntarily consumed morphine for four weeks. Animals drinking only water were used as controls. (**C**) Trajectory plot of rats drinking morphine and water showing entries to the center of the open field. The open field test was conducted during the first hour of light. Data are expressed as means ± SEM. *n* = 8 in each experimental group. * *p* < 0.05; *** *p* < 0.001, two-tailed Student *t*-test.

**Figure 5 ijms-23-03874-f005:**
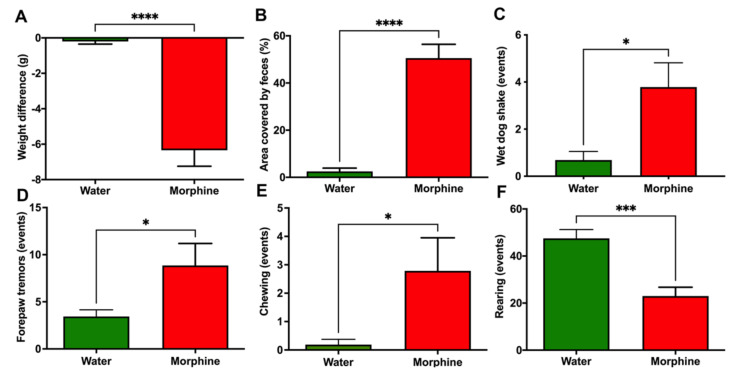
A clear opioid deprivation syndrome is induced by naloxone administration in Wistar rats that had voluntarily consumed morphine. (**A**) Weight loss, (**B**) Area covered by feces (%), (**C**) Wet-dog shake events, (**D**) Forepaw tremors, (**E**) Chewing, and (**F**) Rearing events. Rats that had voluntarily ingested a morphine solution (15–18 mg/kg/day) for four weeks were intraperitoneally injected with 5 mg/kg of the opioid receptor antagonist naloxone. Animals drinking only water were used as controls. Data were recorded for 30 min after naloxone administration and expressed as mean ± SEM. *n* = 8 in each experimental group. * *p* < 0.05; *** *p* < 0.001; **** *p* < 0.0001, two-tailed Student *t*-test.

**Figure 6 ijms-23-03874-f006:**
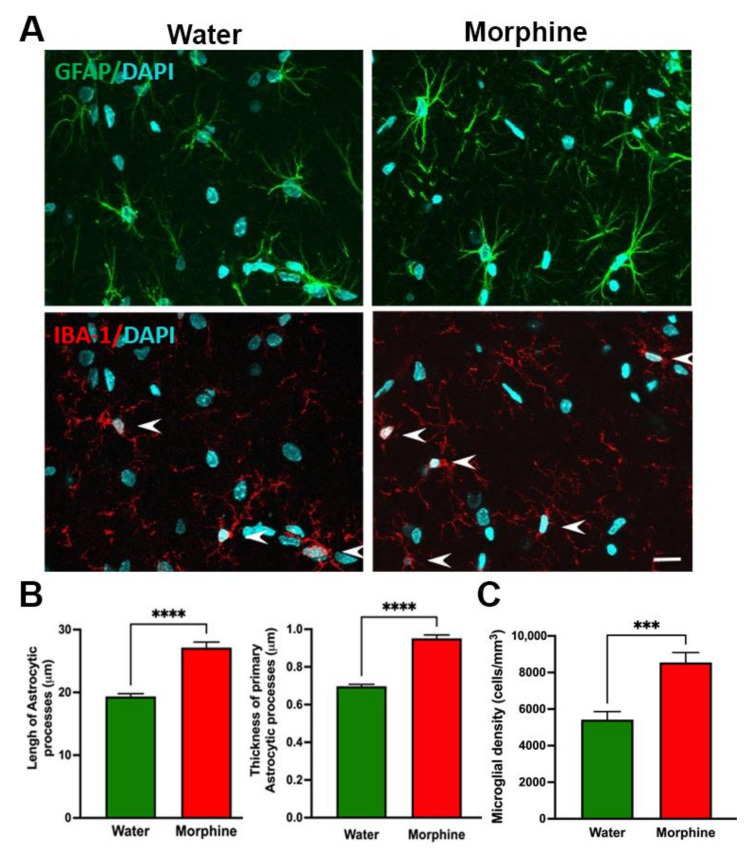
Voluntary morphine intake induces hippocampal neuroinflammation in Wistar rats. (**A**) Representative confocal microphotographs of hippocampal astrocyte GFAP immunoreactivity (green, top) and microglial density (Iba-1, red, shown by arrows, bottom) double-labeling immunoreactivity of rats which had voluntarily drank morphine solution (15–18 mg/kg/day) for four weeks. Animals drinking only water were used as controls. Nuclei were counterstained with DAPI (blue, nuclear marker), scale bar 25 mm. (**B** left) Quantification of total length and (**B** right) thickness of primary astrocytic process and (**C**) Quantification of microglial density evaluated by confocal microscopy and FIJI image analysis software. Data are expressed as mean ± SEM. *n* = 8 in each experimental group. *** *p* < 0.001, **** *p* < 0.0001, two-tailed Student *t*-test.

**Figure 7 ijms-23-03874-f007:**
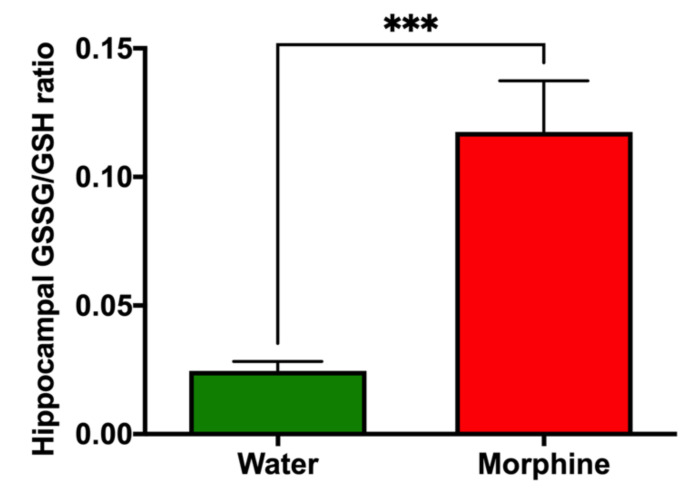
Voluntary morphine intake induces hippocampal oxidative stress in Wistar rats. Hippocampal oxidized/reduced glutathione ratio (GSSG/GSH) in rats that had voluntarily ingested morphine for four weeks. Animals drinking only water were used as controls. Data are expressed as mean ± SEM. *n* = 8 in each experimental group. *** *p* < 0.001, two-tailed Student *t*-test.

**Figure 8 ijms-23-03874-f008:**
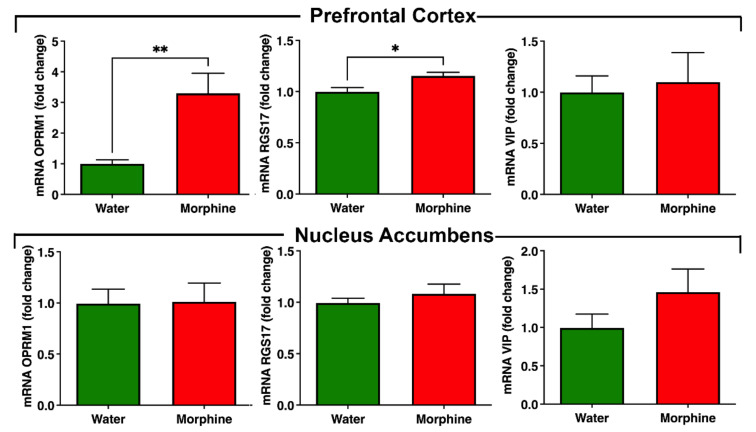
Voluntary morphine intake induces an increase in opioid receptor mRNA levels in pre-frontal cortex but not in nucleus accumbens of Wistar rats. mRNA levels of the morphine receptor OPRM1 and the accessory proteins RGS17 and VIP were determined by RT-qPCR in pre-frontal cortex and nucleus accumbens of rats that had voluntarily ingested morphine solutions for four weeks. Animals drinking only water were used as controls. Expression levels were determined by the ∆∆CT method and normalized against the mRNA levels of the housekeeping genes GAPDH and Hprt1 in the same sample. Data are expressed as mean ± SEM. *n* = 8 in each experimental group. * *p* < 0.05; ** *p* < 0.01, two-tailed Student *t*-test.

## Data Availability

Data are available upon request.

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
