# Peer review of "A Novel Morphine Drinking Model of Opioid Dependence in Rats"

_ijms, 2022, doi:10.3390/ijms23073874_

Round 1
Reviewer 1 Report
This manuscript reported a two-bottle choice animal model of oral morphine self-administration. To exclude the effect of bitter taste, rats were treated to overcome the taste aversion by use of adulterant, quinine. The rat model of morphine self-administration is meaningful for the evaluation of therapeutic interventions for the treatment of morphine dependence. And the model is more reliable since the taste aversion is excluded.
Following points should be addressed:
- Figure 6: the pictures of figure 6A sound from same slices, same views with different staining. However, the method’s statement was not provided. Thus, the results are not reliable.
- Methods: immunostaining, mRNA measurements were not descripted clearly. For example, antibodies for immunostaining, sequence of probes for RT-PCR, etc.
Author Response
Reviewer #1
Rev#1-Comment 1: This manuscript reported a two-bottle choice animal model of oral morphine self-administration. To exclude the effect of bitter taste, rats were treated to overcome the taste aversion by use of adulterant, quinine. The rat model of morphine self-administration is meaningful for the evaluation of therapeutic interventions for the treatment of morphine dependence. And the model is more reliable since the taste aversion is excluded.
Rev#1-Reply 1: Many thanks for your kind comment.
Rev#1-Modification 1: No modification needed.
Rev#1-Comment 2: Figure 6: the pictures of figure 6A sound from same slices, same views with different staining. However, the method’s statement was not provided. Thus, the results are not reliable.
Rev#1-Reply 2: Many thanks for this comment. We agree with the reviewer in the fact that the methodology for the immunofluorescence was not fully described. In this experiment, we performed double-labeling immunofluorescence against the astrocyte marker glial fibrillary acidic protein (GFAP, green) and the microglial marker ionized-calcium-binding adaptor molecule 1 (Iba-1, red) in the same slices. Thus, as the reviewer indicated, these are same views with different staining. This issue is clarified in the legend of figure 6.
Rev#1-Modification 2: A complete description of the methodology used for the double-labeling immunofluorescence in now incorporated in the Material and Methods section (pages 15 and 16, lines 491-507), and in the legend of figure 6 (page 8, line 242).
Rev#1-Comment 3: Methods: immunostaining, mRNA measurements were not described clearly. For example, antibodies for immunostaining, sequence of probes for RT-PCR.
Rev#1-Reply 3: Many thanks for this comment. We agree.
Rev#1-Modification 3: A more detailed description of the methodology used for immunostaining and mRNA measurements, incorporating the sequence of probes for RT-qPCR is added in Material and Methods section (pages 15 and 16, lines 491-507 and pages 16 and 17, lines 533-549).
Reviewer 2 Report
Berrios-Carcamo and colleagues evaluated a new morphine drinking model of opioid dependence in rats. The work is really interesting and the authors provide robust data to validate this animal model for voluntary morphine consumption. The manuscript is well organized and well written. However, there is one aspect that the authors should explain in detail.
The expression levels of OPRM1, RGS17 and VIP expression levels was performed by RT-PCR using beta-actin as reference gene. However, long-term exposition to morphine is known to induce neuroplastic changes in the brain, including morphological changes in neurons of the cortex, nucleus accumbens, and caudate putamen. Since actin is an important protein of the neuron cytoskeleton and is involved in their remodeling processes, it does not seem to be the most appropriate gene to use in normalization. I strongly suggest the authors consider re-evaluating their data with a different reference gene, such as GDPH. Indeed, the absence of changes in OPRM1 expression in accumbens are quite difficult to understand in view of previous reports and could be due to a wrong analysis of PCR experiment.
Author Response
Reviewer #2
Rev#2 Comment 1: Berrios-Carcamo and colleagues evaluated a new morphine drinking model of opioid dependence in rats. The work is really interesting, and the authors provide robust data to validate this animal model for voluntary morphine consumption. The manuscript is well organized and well written.
Rev#2 Reply 1: Many thanks for your kind comment.
Rev#2 Modifications1: No modification needed.
Rev#2 Comment 2: The expression levels of OPRM1, RGS17 and VIP were performed by RT-PCR using beta-actin as reference gene. However, long-term exposition to morphine is known to induce neuroplastic changes in the brain, including morphological changes in neurons of the cortex, nucleus accumbens, and caudate-putamen. Since actin is an important protein of the neuron cytoskeleton and is involved in their remodeling processes, it does not seem to be the most appropriate gene to use in normalization. I strongly suggest the authors consider re-evaluating their data with a different reference gene, such as GDPH. Indeed, the absence of changes in OPRM1 expression in accumbens are quite difficult to understand in view of previous reports and could be due to a wrong analysis of PCR experiment.
Rev#2 Reply 2: Many thanks for this comment. We fully agree with the reviewer. The choosing of the correct housekeeping gene in different target tissues is a difficult task. Following reviewer’s suggestion we re-analyzed our RT-qPCR data using GAPDH as a housekeeping gene but also hypoxanthine phosphoribosyltransferase 1 (Hprt1), since this gene has been proposed as a reliable housekeeping gene for brain tissue (Feria-Romero, I et al, Gene 2021, 769:145255.). Thus, the mRNA level of each target gene was normalized against the geometric mean of the mRNA levels of the housekeeping genes GAPDH and Hprt1 in the same sample.
Rev#2 Modifications 2: Figure 8 was modified to show the expression levels of OPRM1, RGS17 and VIP in prefrontal cortex and nucleus accumbens, using GAPDH and Hprt1 as internal controls for normalization. This information is included in the Material and Methods section (page 16, lines 538-540) and in the legend of figure 8 (page 10, lines 279-280).
Round 2
Reviewer 2 Report
The authors have satisfactorily answered all my questions.